# Gut carriage of antimicrobial resistance genes in women exposed to small-scale poultry farms in rural Uganda: A feasibility study

**Ana A. Weil**[1,2☯¤]**, Meti D. Debela**[1☯]**, Daniel M. Muyanja**[3]**, Bernard Kakuhikire**[3]**, Charles Baguma**[3]**, David R. Bangsberg**[3,4]**, Alexander C. Tsai** [2,3,5,6]**, Peggy S. Lai** [1,2,7] *

**1** Department of Medicine, Massachusetts General Hospital Boston, Boston, Massachusetts, United States of America, **2** Harvard Medical School, Boston, Massachusetts, United States of America, **3** Mbarara University of Science and Technology, Mbarara, Uganda, **4** Oregon Health & Science University, Portland State University School of Public Health, Portland, Oregon, United States of America, **5** Harvard Center for Population and Development Studies, Cambridge, Massachusetts, United States of America, **6** Center for Global Health and Mongan Institute, Massachusetts General Hospital, Boston, Massachusetts, United States of America, **7** Harvard T.H. Chan School of Public Health, Boston, Massachusetts, United States of America

☯ These authors contributed equally to this work.
¤ Current address: Department of Medicine, Division of Allergy and Infectious Diseases, University of Washington, Seattle, Washington, United States of America
* pslai@hsph.harvard.edu

**Data Availability Statement:** All relevant data are within the manuscript and its Supporting Information files.

## Abstract

### Background

Antibiotic use for livestock is presumed to be a contributor to the acquisition of antimicrobial resistance (AMR) genes in humans, yet studies do not capture AMR data before and after livestock introduction.

### Methods

We performed a feasibility study by recruiting a subset of women in a delayed-start randomized controlled trial of small-scale chicken farming to examine the prevalence of clinically-relevant AMR genes. Stool samples were obtained at baseline and one year post-randomization from five intervention women who received chickens at the start of the study, six control women who did not receive chickens until the end of the study, and from chickens provided to the control group at the end of the study. Stool was screened for 87 clinically significant AMR genes using a commercially available qPCR array (Qiagen).

### Results

Chickens harbored 23 AMR genes from classes found in humans as well as additional vancomycin and β-lactamase resistance genes. AMR patterns between intervention and control women appeared more similar at baseline than one year post randomization (PERMANOVA $R^2 = 0.081$, $p = 0.61$ at baseline, $R^2 = 0.186$, $p = 0.09$ at 12 months) Women in the control group who had direct contact with the chickens sampled in the study had greater similarities in AMR gene patterns to chickens than those in the intervention group who did not have direct contact with chickens sampled ($p = 0.01$). However, at one year there was a trend

**Funding:** Funding was provided by National Institutes of Health grants K23 ES023700 (PSL), P30 00002, and K23 MH096620 (ACT), and K08AI123494 (AAW) (https://www.nih.gov/), Harvard School of Public Health-National Institute of Environmental Health Sciences and Center for Environmental Health (P30ES000002) Pilot Project Grant (PSL) (https://www.hsph.harvard.edu/niehs/), American Lung Association Biomedical Research Grant RG-346990 (PSL) (https://www.lung.org/), Harvard Catalyst (UL1 TR001102) Early Clinical Data Support Pilot Grant (PSL) (https://catalyst.harvard.edu/), and Friends of a Healthy Uganda (DRB, ACT) (https://attackpoverty.org/locations/friends-of-uganda/). The funders had no role in study design, data collection, analysis, decision to submit the work for publication, or preparation of the manuscript.

**Competing interests:** The authors have declared that no competing interests exist.

towards increased similarity in AMR patterns between humans in both groups and the chickens sampled (p = 0.06).

## Conclusions

Studies designed to evaluate human AMR genes in the setting of animal exposure should account for high baseline AMR rates. Concomitant collection of animal, human, and environmental samples over time is recommended to determine the directionality and source of AMR genes.

## Trial registration

ClinicalTrials.gov Identifier NCT02619227.

## Introduction

Antimicrobial resistance (AMR) is a global public health crisis. Although estimates vary on the severity of the problem, one report has suggested that by 2050, 10 million deaths a year worldwide will be attributed to antimicrobial resistance [1], with crude estimates of the annual economic costs totaling 55 billion dollars in the United States alone [2]. This problem may be accentuated in resource-limited settings due to a likely higher burden of infectious disease, little to no antimicrobial stewardship, less resources for microbiology testing, possible limited access to antibiotics targeting highly resistant pathogens, insufficient sanitation and hygiene infrastructure for managing human and animal waste. Although prior AMR studies have focused on hospitalized patients and recent administration of antimicrobials to treat infections, an updated view of AMR as a public health problem has highlighted the importance of AMR as a "One Health" problem; that is, viewing human, animal, and environmental health as interconnected and interdependent [3–5]. Antibiotics are widely used in livestock farming to enhance animal health and increase productivity [6], and this practice is thought to be one contributor to the problem of AMR among humans. However, most available studies are cross-sectional and/or focused on single organisms or pathogens [7–9], and these study designs lack the ability to determine causality. More robust study designs are needed to determine the effect size that antimicrobials used in livestock farms has on transmission of AMR genes to humans [10].

Surveillance data in 2005 showed that livestock production in Uganda accounted for about 5% of total Ugandan gross domestic product [11], with an estimated annual production of 70.8 million total livestock including cattle, pigs, sheep and goats, and poultry [12]. Studies of poultry farms in Uganda have identified multiple mechanisms of AMR in *Escherichia coli* strains isolated from healthy chickens [13, 14], suggesting that poultry farms may serve as a reservoir of AMR genes for humans. Few studies have evaluated how the initiation of chicken farming relate to AMR in humans, partly due to difficulty in obtaining pre-intervention samples for AMR testing. In this study we determined the feasibility of a longitudinal study of AMR gene patterns in a subset of participants enrolled in a delayed intervention RCT of small-scale chicken farms in rural Uganda.

## Materials and methods

### Study design and study population

We recruited participants from an existing randomized clinical trial (RCT) of small-scale chicken farming (ClinicalTrials.gov Identifier: NCT02619227) [15]. Participants were chosen

by convenience sample from the original trial and assessment of AMR gene carriage was added as a post-hoc aim. In the primary waitlist-controlled RCT conducted in 2015, 92 women living in Mbarara, Uganda were recruited and randomized to receive training, raw materials, and broiler hybrid chicks either immediately (intervention group), or after at least a 12-month delay (control group). Chicken coops were constructed to house chicks as part of the study protocol, and study participants were the primary caretakers for the broilers. Chicks, feed, and medications were given to intervention participants as a series of escalating microloans starting from 15, 50, then 100 chicks with loans paid back prior to the receipt of the next batch of chicks (see S1 Fig). Broiler chicks were sourced from a single distributor based in Kampala, Uganda and underwent a standard care protocol by participants during the brooding period which lasted approximately 8 weeks. Under supervision, participants administered vaccines to the chicks against Newcastle, Gumboro, fowl typhoid, and fowl pox. Participants also routinely administered dietary supplements to chicks in their drinking water during the brooding period as part of a protocol to boost growth. This included two oxytetracycline-containing medications; Alamycin chick formula given for the first 2–3 weeks of brooding and Oxiveto given weekly for four weeks. In addition, Coccid (which contains amprolium) was given once weekly for four weeks to prevent coccidiosis. Chicken feed was sourced from a single distributor based in Mbarara, Uganda. Routine surveys were administered as part of the RCT to monitor behaviors such as recent antimicrobial use (in both chickens and humans) and vaccination status in chickens, and data pertinent to this study was extracted from the survey developed for the larger RCT (see Supplement). The primary trial was designed as a series of microloans in the form of chickens. Per study protocol, the time from chick acquisition to slaughter was 8 weeks, although participants were given the option of an additional two weeks to sell their chickens and pay back the loan.

The timing of stool sample collection is depicted in S1 Fig. Stool samples from six chicken coops belonging to the 6 control participants were collected by retrieving fresh chicken stool once at approximately 18 months after randomization, between 20 and 41 days after the control group had received their chickens as part of the delayed-start randomized controlled trial design. Human samples could not be collected at the 18 month timepoint when chicken samples were collected. The 6 control participants were chosen based on participants who had stool samples collected from the chickens, and the 5 intervention participants lived in the same villages as the control participants. Stool sample collection was added post-hoc for this feasibility study as an addition to the original study. At baseline, before chickens were introduced into the intervention households and at 12-month follow up after chicken introduction in the intervention group, we obtained fresh stool samples from participants during research clinic visits. Stool samples were frozen within one hour of collection in generator-backed -80˚C freezers in the research laboratories of the Mbarara University of Science and Technology. All samples were subsequently transported on dry ice to Massachusetts General Hospital for further processing. All study procedures were approved by the Research Ethics Committee of Mbarara University of Science and Technology (Protocol #30/11-14) and the Partners Human Research Committee (Protocol #2015P000227/BWH). Consistent with national guidelines, we also received clearance for the study from the Ugandan National Council of Science and Technology (Protocol #HS 1746) and the President's office.

**Sample processing, AMR gene identification and quantification.** Microbial DNA was extracted from 100mg of chicken and human stool samples, and from a reagent-only negative control using the PowerSoil DNA extraction kit (Qiagen, Valencia, CA) according to the manufacturer's instructions. The presence of AMR genes was screened using a commercially available AMR gene identification microbial DNA polymerase chain reaction (PCR) array (Qiagen, Valencia, CA, cat. No. 330261) according to the manufacturer's instructions. This array targets

six major classes of antibiotics (aminoglycoside, β–lactam, erythromycin, fluoroquinolone, macrolide–lincosamide–streptogramin B, tetracycline, and vancomycin) and includes genes with multi-resistance potential. Briefly, 500ng template microbial DNA was mixed with 1275 µl qPCR mastermix (Qiagen) and nuclease-free water was added to reach a final volume of 2550 µl. 25 µl of reaction mix was added to a 96-well PCR plate containing a pre-dispensed mixture of lyophilized primers and probes for each of the 87 AMR genes. qPCR was performed using Applied Biosystems 7500 Fast Real-Time PCR System using thermal cycling conditions of initial denaturation at 95˚C for 10 minutes, followed by 40 cycles of denaturation at 95˚C for 15 seconds, and annealing at 60˚C for 2 minutes. Raw cycle threshold (CT) values were analyzed using the Microbial DNA qPCR Array data analysis template. One replicate per sample was tested. The efficiency of the PCR instrument and the quality of mastermix were determined by measuring the CT for the control sample between 20 and 24. Validity of the control ensured that potential PCR inhibitors in the sample did not interfere with measurements. A no-template and nuclease-free control were also included to evaluate for the presence of laboratory based contaminants.

**Data analysis, visualization and statistical analysis.** Determination of detection of AMR genes was performed according to the manufacturer's (Qiagen) guidelines, described here in brief. The presence or absence of each AMR gene was determined as follows: present if $\Delta CT > 6$, not detected if $\Delta CT < 3$, and inconclusive if $\Delta CT$ was $\geq 3$ and $\leq 6$. To visualize the results of AMR gene presence or absence in each sample, we created a heatmap using the *ggplot2* R package [16]. In order to visualize global patterns of AMR genes over time in the human samples and difference between the chicken samples, we chose to use the Jaccard dissimilarity index. Briefly, the Jaccard index calculates the proportion of unshared features (here AMR genes) out of the total number of features (here AMR genes) recorded between any two samples, an approach used in other studies of high-dimensional antimicrobial resistance data [17, 18]. To calculate the Jaccard index, we first created a sample by feature matrix denoting the presence or absence of each AMR gene in each sample. Presence/indeterminacy/absence were determined using the $\Delta CT$ method described above according to manufacturer recommendations, with the following value assignments; present = 1, indeterminate = 0, absent = 0. Visualization of the dissimilarities in AMR gene patterns was performed using the plot_ordination() function as implemented in the *phyloseq* R package [19]. To test the hypothesis that AMR gene patterns in human control groups were the same or different at baseline and one year post-randomization, we performed permutational multivariate analysis of variance (PERMANOVA) [20, 21] on the Jaccard index with 10,000 permutations as implemented in the *vegan* R package [22]. To determine the similarities in AMR gene patterns between human control and intervention groups over time compared to chickens, we computed the distance between the Jaccard index of each sample to the centroid of all chicken samples [23]. In this plot, a shorter distance between data points indicates increased similarity in AMR gene patterns. Measurements were calculated using the dist_between_centroids() function implemented in the *usedist* R package [24]. For statistical testing, we performed a mixed effects model as implemented in the mgcv R package [25] where the outcome was the calculated distance between each sample and the centroid of the chicken samples, covariates were group membership (intervention vs control) and time (baseline vs follow-up), adjusting for repeated measures in a subject. All statistical analyses were performed in the R programming language [26]. Two-sided p values of $< 0.05$ were considered statistically significant.

## Results

We collected stool from five women in the intervention group and six women in the control group, from 11 separate households in Nyakabare parish, Mbarara district, Uganda. Mbarara

**Table 1. Baseline characteristics of study participants.**

|  | Control | Intervention |
|---|---|---|
| n | 6 | 5 |
| Age, years | 40 [34–43] | 33 [25–40] |
| Farming | 6 (100%) | 5 (100%) |
| Antibiotic use in prior three months |  |  |
| At 0 months | 1 (17%) | 0 (0%) |
| At 12 months | 1 (17%) | 1 (20%) |
| Animal contact | 5 (83%) | 5 (100%) |
| Village chickens[a] | 2 (33%) | 5 (100%) |
| Cows | 2 (33%) | 2 (40%) |
| Goats | 4 (67%) | 4 (80%) |
| Pigs | 1 (17%) | 0 (0%) |
| Dogs | 2 (34%) | 2 (40%) |
| Cats | 2 (34%) | 2 (40%) |

[a]Village chickens refer to free-range chickens that do not receive vaccinations or medications, and do not require an enclosure.

is located in a rural area of Uganda approximately 260km southwest of Kampala, the capital city. The local economy is largely dominated by animal husbandry, petty trading, subsistence agriculture, and supplemental migratory work. Food and water insecurity are common [27–29]. In this study, samples were collected between August 11, 2015 and June 8, 2017. The median age of participants was 35 years, and self-reported demographic data are listed in Table 1. All participants were women involved in subsistence farming. At baseline, 10 of the participants reported regular animal contact and a minority reported recent antibiotic use.

## AMR genes detected

Stool samples from chickens, and from pre- and post-intervention human control and intervention groups were assayed for AMR genes using a validated quantitative polymerase chain reaction (qPCR) assay. All of the no-template controls and positive PCR controls passed the quality control thresholds determined by the manufacturer (S1 Table). At baseline, the stool of study participants in both control and intervention groups harbored β-lactamase, aminoglycoside, fluoroquinolone, macrolide and tetracycline AMR genes found in the stool (Table 2). Seven new AMR genes were detected after one year in the intervention group, and four of these were present in chickens (SHV, SHV[238G240E], QnrS, QnrB-5 group). Six new AMR genes were detected after one year in the control group, and one of these was present in chickens (CTX-M-1 group). Overall, AMR genes were detected from five classes of antimicrobials in humans, and six classes in chickens.

## AMR gene class trends between groups and over time

During the study period there was an overall increase in AMR genes in both the control and intervention groups. The most prevalent AMR genes were *tetA* and *tetB*, which confer tetracycline efflux pumps, and these were found in all chickens tested. *tetA* and *tetB* were also found in the majority of human participants in the study at baseline and follow-up timepoints, as shown in a heatmap of our overall results (Fig 1 and S1 Table). β-lactamases were also highly prevalent in both humans and chickens, with Class A and C β-lactamase AMR genes found in

**Table 2. Antimicrobial resistance (AMR) genes in participants detected at baseline and one year post-intervention, and in chickens.** Among study participants, newly detected genes after one year are shown in bold. Baseline grouping includes both intervention and control group participants. AMR gene detection was measured using a qPCR array (Qiagen). Raw cycle threshold (CT) values were used to determine detection of AMR, defined as positive if ΔCT >6, not detected if ΔCT <3 and inconclusive if ΔCT was ≥ 3 and ≤6, as per the manufacturer's instructions. Raw qPCR data is shown in S1 Table. Gene names are italicized and names of gene classes are not.

| Antibiotic classification | Women at baseline (0 months) | Women in intervention group (12 months) | Women in control group (12 months) | Chickens (18 months*) |
|---|---|---|---|---|
| Aminoglycoside resistance | *aadA1* | ***aacC2***, *aadA1* | ***aacC2***, *aadA1* | *aadA1* |
| Class A β-lactamase | CTX-M-1 group, CTX-M-9 group, SHV, SHV(156G), SHV(238G240E) | CTX-M-1 group, **SHV**, SHV(156G), **SHV (238G240E)** | **CTX-M-1 group**, SHV, **SHV (156D)**, SHV(156G), SHV (238G240E) | CTX-M-1 group, SHV, SHV(156G), SHV(238G240E), SHV(238S240E), SHV(238S240K) |
| Class B β-lactamase | *ccrA* | *ccrA* | - - | *ccrA* |
| Class C β-lactamase | ACT-1 group, ACT 5/7 group, *ACC-3*, MIR | ACT-1 group, **ACT 5/7 group**, MIR | ACT-1 group, ACT 5/7 group, ***CFE-1***, **LAT**, MIR | ACT-1 group, MIR |
| Class D β-lactamase | - - | - - | - - | OXA-10 group, OXA-58 group |
| Fluoroquinolone resistance | QnrS, QnrB-1 group, QnrB-5 group | ***AAC(6)-Ib-cr*, QnrS, QnrB-5 group** | ***AAC(6)-Ib-cr***, QnrS, QnrB-1 group, QnrB-5 group | QnrS, QnrB-5 group, QnrB-8 group |
| Macrolide Lincosamide Streptogramin_b | *ermB*, *mefA* | *ermB*, *mefA* | *ermB*, *mefA* | *ermA*, *ermB*, *ermC*, *mefA*, *msrA* |
| Tetracycline efflux pump | *tetA*, *tetB* | *tetA*, *tetB* | *tetA*, *tetB* | *tetA*, *tetB* |
| Vancomycin resistance | - - | - - | - - | vanB, vanC |

* Chicken stool was collected 18 months after randomization, but between 20–41 days after chick delivery to the control group.

humans at both baseline and follow-up timepoints, regardless of chicken exposure, and the Class C β-lactamase MIR present in nearly all study participants. However, Class D β-lactamases were found only in chickens. AMR genes in the Class C β-lactamase group, which includes the clinically important *ampC* β-lactamases responsible for inducible resistance upon exposure to specific antibiotics were particularly dynamic over time, with two AMR genes emerging in the control group after one year that were not seen in other groups (*CFE-1* and LAT), and the loss of *ACC-3*, which was found in the baseline population and not detected upon follow up [30]. Fluoroquinolone and macrolide resistance were widespread over all groups and timepoints. Chicken AMR genes detected included two vancomycin resistance genes that were not found in humans.

To address the question of whether AMR gene profiles were different between intervention and control groups at baseline and 12 months after randomization, we used permutational analysis of variance (PERMANOVA) on the Jaccard distance to determine whether the centroids of the intervention and control groups differ at baseline and 12 months after randomization in the control vs. intervention groups. At baseline, there was no difference in AMR resistance patterns between control and intervention groups (PERMANOVA $R^2$ = 0.081, p = 0.61), whereas at 12 months there was a trend towards a difference in AMR patterns between intervention and control groups (PERMANOVA $R^2$ = 0.186, p = 0.09). We used an ordination plot to depict patterns of AMR gene composition across groups and over time (Fig 2).

To determine the similarity of AMR gene patterns of human samples compared to the chicken samples, we computed the distance between the Jaccard index of each sample to the centroid of the chicken samples (Fig 3). A shorter distance between data points indicates increased similarity in AMR gene pattern with the chicken samples, while a higher distance indicates decreased similarity in AMR gene pattern with the chicken samples. To identify predictors of similarity between human and chicken AMR patterns, we used mixed effects models where the outcome was the distance between each human sample compared to the centroid of

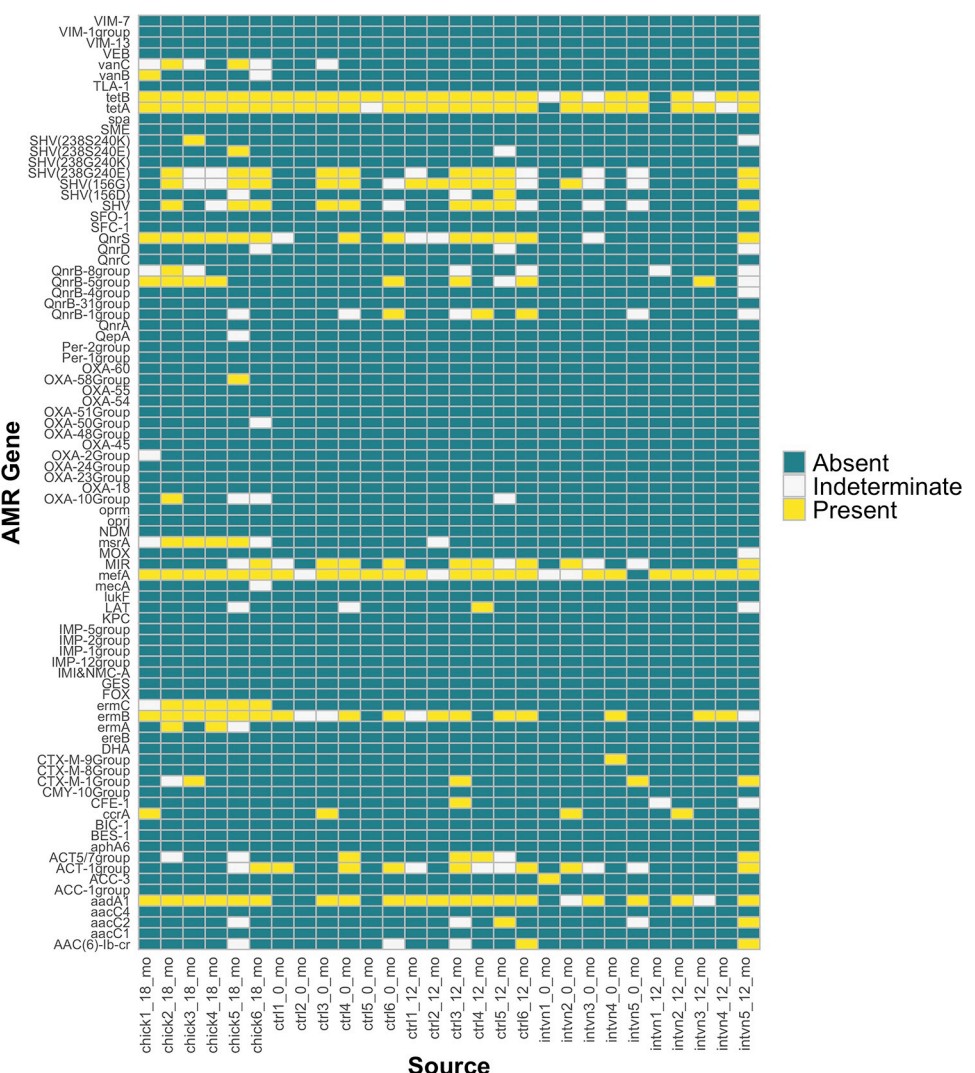

**Fig 1. Heatmap demonstrating whether antimicrobial resistance (AMR) genes were present, absent, or indeterminate in human and chicken samples at different timepoints.**

the chicken samples, with predictors being control vs. intervention group and timepoint (baseline vs. 12 months), adjusting for repeated measures in a person. The AMR gene pattern of the control group is more similar to the AMR gene pattern in their chickens than the intervention group to the control group's chickens (b = 0.128, p = 0.014, intervention vs. control group; Note *more* positive b indicates *less similarity* with chicken samples). There was a trend towards increasing similarity of AMR gene patterns between all human groups (control and intervention) and chickens at one year compared to baseline though it did not reach statistical significance (b = -0.067, p = 0.059, 12 month vs baseline). In this latter comparison, the effect size was negative, which is consistent with increased similarity between follow-up AMR gene patterns in both the intervention and control groups compared to chicken AMR gene patterns.

## Discussion

In this study, we find that tetracycline-exposed chickens and humans who care for them harbor AMR genes from multiple gene classes. Over one year, AMR gene carriage increased in all

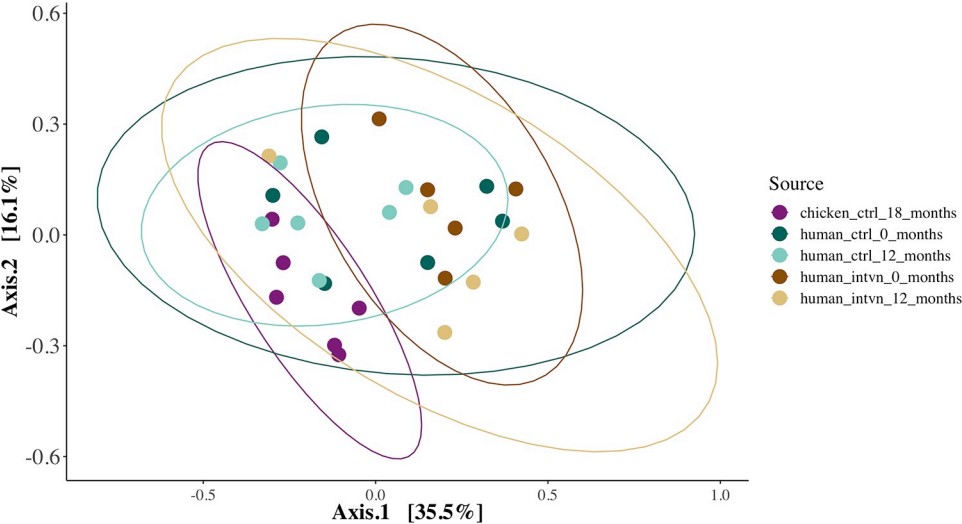

**Fig 2. Ordination plot of the Jaccard dissimilarity index of AMR gene patterns between groups.** The proportion of unshared AMR genes out of the total number of AMR genes detected between any two samples is shown. More similar samples will appear closer together on the plot. The ellipse depicts the 95% confidence ellipse around each sample group. At baseline, there were no statistically significant differences between AMR gene patterns between intervention and control groups (PERMANOVA $R^2 = 0.081$), whereas at 12 months, there was a trend towards different AMR gene patterns (PERMANOVA $p = 0.09$) between intervention and control groups.

study participants although we only tested two time points in humans. Women who did not care for chickens during the 12 months of human sampling (control group) harbored many of the same AMR genes at one year. There were greater AMR gene pattern similarities between chickens and the humans who had direct interaction with the chickens in the study. This latter finding should be interpreted conservatively and with several caveats: first, chicken stool samples were obtained only after human stool sample collection, and were obtained only from the control group who received chickens at the end of the study. It is possible that the control group samples were more similar to their chickens either due to community-wide human to chicken AMR gene transmission, or because the control group and chickens had a common environmental source such as community wells for water.

Shared gut organisms between animals and humans increase when close contact occurs between groups, such as in animal husbandry [31, 32]. These shared environments can result in transmission events, which range from zoonotic infections to the spread of benign commensal microbes, or events that represent potential harm to humans or animals, such as acquisition of AMR genes. Pathogens resistant to antibiotics result in more severe illness and increased mortality in humans compared with infections caused by susceptible bacteria [33]. Consistent with the One Health concept, we found that humans and chickens with direct contact had greater similarities in AMR gene carriage in the gut, although the directionality of transmission could not be determined based on our study design. Stool samples from the control group were obtained prior to chicken introduction, and thus increased similarity in AMR gene patterns with the chickens in this study who had direct contact with the control group could be due to human to chicken transmission, or a common environmental source. Prior studies have demonstrated that the introduction of tetracycline-supplemented feed to chickens led to increased carriage of multi-drug resistant bacteria in the feces of chickens, and after 3 months, a rise also in resistant intestinal bacteria in farm workers caring for these chickens. Bidirectional transfer of AMR genes is possible, likely through mobile genetic elements. Additionally, we observed overall that there are many AMR genes in chickens in rural Uganda.

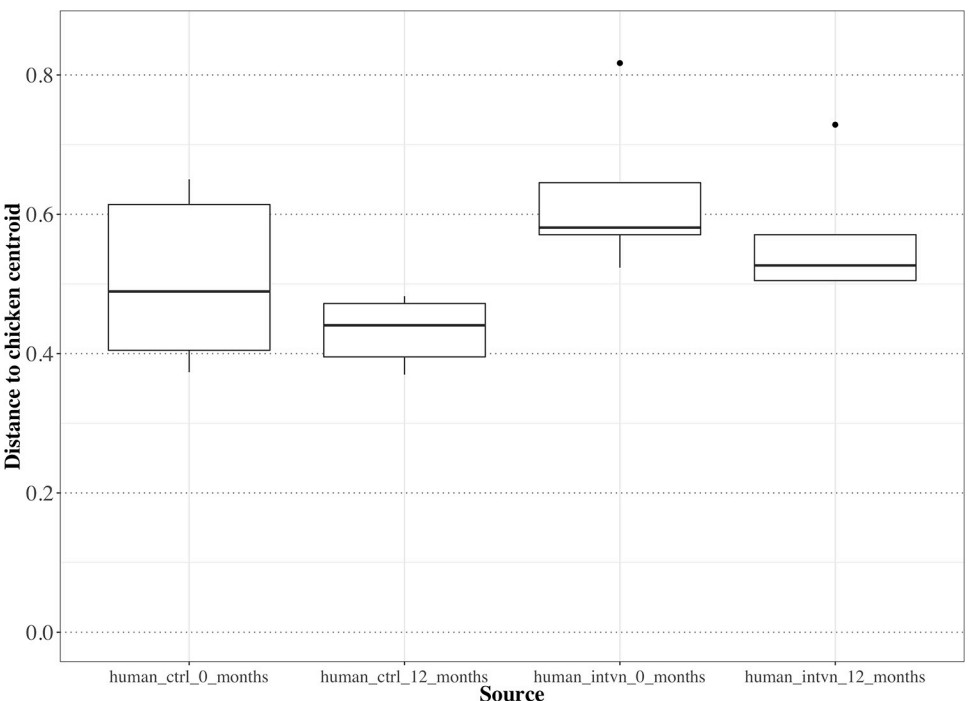

**Fig 3. Boxplot of the distance between sample groups and the centroid of the chicken stool samples based on AMR gene pattern.** To demonstrate the comparison of the AMR gene pattern of each human sample to the chicken samples at baseline and follow up, we computed the distance between the Jaccard index of each sample to the centroid of all chicken samples. Here, a *shorter* distance indicates *increased* similarity in AMR gene pattern of the human sample in relation to the centroid of the chicken samples gene patterns, whereas a *longer* distance indicates *decreased* similarity in AMR gene pattern of that human sample compared to the chicken samples gene patterns. The chicken sample centroid is set at zero. The AMR gene pattern of the chicken samples is more similar to the AMR gene pattern in the control group rather than the intervention group (p = 0.014); note that chicken samples were obtained from the control group. Differences in AMR gene patterns over time did not reach statistical significance (p = 0.059), although at follow-up, the AMR gene patterns in both control and intervention group humans were more similar to AMR gene patterns in chicken samples.

Tetracycline resistance genes are often found to be widespread among livestock treated with antibiotics, including in Africa [34]. Use of tetracycline in livestock has been associated with increased colony counts of tetracycline-resistant human pathogens in treated animals [35]. While tetracycline was the only antibiotic administered to chickens in this study, a wide range of AMR genes from six different classes were detected in chickens. Many β-lactamase AMR genes with direct links to difficult-to-treat human infections were also detected. For example, the CTX-M-1 Group can confer an extended-spectrum beta lactamase (ESBL) phenotype, and is the most commonly found gene in *Escherichia coli* in the few surveys of AMR genes that have been conducted in African livestock [34]. CTX-M-1 was detected after one year in our control group and was also present in chickens in this study,. CTX-M-1 was also present in the intervention group both at baseline and follow up. While the directionality of CTX-M-1 transmission between control participants and chicken exposure cannot be evaluated in this study, our results demonstrate that CTX-M-1 is circulating in this population among chickens and humans. The Carbapenemases OXA-10 and OXA-58 Group were also found in chicken stool in our study and were not found in humans, and confer a concerning degree of antimicrobial resistance [36]. Reasons that these AMR genes have not emerged into the human population are unknown, and may be due to a lack of selective pressure (ie chickens and humans not yet exposed to carbapenem antibiotics) at the time of our study. Similarly, the

VanB and VanC genes found in chickens are known to confer vancomycin (glycopeptide antibiotic) resistance to Enterococci, a common genus of the colonic flora, resulting in the clinically important vancomycin resistant enterococcus (VRE). Avoparcin, an antibiotic also from the glycopeptide class, was widely used in livestock and poultry in Europe and linked to VRE isolates in animals. This drug was outlawed for use in animals in the European Union in 1997, although VRE isolates have persisted in some poultry populations after use ceased [37, 38]. Although Avoparcin was not known to be administered to the chickens in this study, it is sold in Uganda as a livestock supplement.

In the humans we studied, numerous additional AMR genes were detected in both the intervention and control groups at the one year follow up timepoint compared to baseline. There are several possible explanations for this finding. First, it is possible that the higher amount of AMR gene content at the one year follow up is simply due to ongoing, transient fluctuations in AMR presence that occur over time. Another possibility is that the population may be trending toward increased AMR gene content over relatively short time periods, and that AMR genes in this population are widespread and dynamic. For example, in the fluoroquinolone class, the *AAC(6)-lb-cr* gene is often found on a multiresistance plasmid with other AMR genes, and this gene was detected in both human groups after follow up and was not found in chickens. This indicates that the increased AMR gene patterns over time seen in this population may also originate from sources unrelated to chicken exposure, such as environmental sources [39]. Over one year, we observed that microbial community profiles in humans were significantly altered with (intervention group) or without chicken exposure (control group). We also note shared AMR genes between humans and chickens. Possible explanations for our findings could be that 1) there is a common source of AMR genes in both chickens and humans, for example environmental sources such as water; 2) the possibility exists that AMR genes may be transmitted from humans to chickens; 3) our data does not allow us to comment on transmission of AMR genes from chickens to humans as chicken stool samples were collected after the human stool samples. However, all chicks were from the same distributor and underwent the same care protocol and thus it is possible that some of the AMR genes acquired by humans over time were from direct or indirect chicken contact.

In this study, we describe point prevalence estimates of AMR genes over two timepoints in humans. Our study has some limitations. This pilot study does not evaluate for the directionality or source of transmission of AMR genes detected in humans and chickens, because chicken stool samples were not collected at the same timepoints, and only two human timepoints were collected. Larger longitudinal studies should include repeated measures within subjects with assessments of correlation within subjects and modeling assessments of associations to account for repeated measures and autocorrelation. This limited our ability to detect random variation from true trends. We also did not assess cross-contact between enrolled participants within and between villages. In this study we did not aim to assess AMR transmitted from other community sources, and focused only on AMR gene content in chickens and humans. Additionally, our sample size was small, and our detection method often identified gene classes, preventing us from commenting on presence of specific genes. Additionally, the number of replicates collected over time in both humans and chickens are unlikely to capture the diversity of timing relative to chicken production cycle and antibiotic use in that cycle. Our qPCR detection of genes was conducted with single replicates due to the cost of the AMR gene arrays. Despite these limitations, our study does highlight the prevalence of circulating AMR genes in people and in chickens in a rural Ugandan population. Our results offer practical design suggestions for future studies evaluating AMR gene transmission in animal husbandry settings. Based on our experience, we would recommend measurement of a wide range of AMR genes at several timepoints, since at baseline a significant number of AMR genes were already

present in humans. Sampling from humans, livestock, as well as shared environmental samples (such as water sources) would be required to establish patterns of temporal transmission. A randomized controlled trial design for livestock exposure, as well as molecular evaluation of genetic similarity between bacterial strains harboring AMR genes will be critical for evaluating causality and directionality of transmission.

The World Health Organization Expert Guidelines Development Group tasked with addressing the worldwide crisis of increasing AMR recommend complete restriction of all classes of medically important antibiotics in food-producing animals for growth promotion [40]. Although this was issued as a strong recommendation, evidence to support the recommendation was deemed "low-quality" due to a lack of supportive studies. Here, we describe changes in the AMR gene profile in stool of humans over two timepoints, and highlight the prevalence of AMR genes in both humans and livestock in a rural Ugandan population. In future studies, to confirm the suspected epidemiologic links that may be responsible for the results in this pilot study, genotyping methods to define mobile elements and strain-specific analysis of AMR genes found in humans exposed to antibiotic-treated livestock are needed. A randomized trial design where simultaneous acquisition of human, livestock and environmental samples with a high frequency of sampling may be useful to define susceptibility factors for acquisition of AMR genes.

## Supporting information

**S1 Fig. Study design overview.**
(JPG)

**S1 Table. Cycle threshold (CT) values of AMR detection from human and chicken stool samples and controls used in this study.** PPC = Positive PCR Control.
(XLSX)

**S1 Method.**
(DOCX)

## Acknowledgments

The authors thank the participants in this study. In addition to the named study authors, HopeNet Study team members who contributed to data collection and/or study administration during all or any part of the study were as follows: Phiona Ahereza, Owen Alleluya, Gwendoline Atuhiere, Patience Ayebare, Dickson Beinomugisha, Bridget Burns, Augustine Byamugisha, Patrick Gumisiriza, Clare Kamagara, Allen Kiconco, Noel Kansiime Kiza, Viola Kyokunda, Moran Mbabazi, Amy McDonough, Juliet Mercy, Patrick Lukwago Muleke, Elijah Musinguzi, Moran Owembabazi, Sarah Nabachwa, Elizabeth Betty Namara, Immaculate Ninsiima, Mellon Tayebwa, Specioza Twinamasiko, and Dagmar Vořechovská. Clean Air Study team members who contributed to data collection and/or study administration during all or any part of the study were as follows: Solome Kobugyenyi, Alex Mutungi, John Bosco Tumuhimbise, Joy Namara Karakire, Collins Agaba. A preliminary analysis of these data was presented at IDWeek, San Francisco, California, USA, October 5, 2018.

## Author Contributions

**Conceptualization:** Ana A. Weil, Peggy S. Lai.

**Data curation:** Peggy S. Lai.

**Formal analysis:** Peggy S. Lai.

**Funding acquisition:** Peggy S. Lai.

**Investigation:** Ana A. Weil, Meti D. Debela, Daniel M. Muyanja, Bernard Kakuhikire, Charles Baguma, David R. Bangsberg, Alexander C. Tsai, Peggy S. Lai.

**Resources:** Peggy S. Lai.

**Supervision:** Peggy S. Lai.

**Visualization:** Peggy S. Lai.

**Writing – original draft:** Ana A. Weil, Peggy S. Lai.

**Writing – review & editing:** Ana A. Weil, Meti D. Debela, Daniel M. Muyanja, Bernard Kakuhikire, Charles Baguma, David R. Bangsberg, Alexander C. Tsai, Peggy S. Lai.

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
