## [Decision Letter · Decision Letter 0]

11 Mar 2020

PONE-D-20-03262

Gut carriage of antimicrobial resistance genes in women exposed to small-scale poultry farms in rural Uganda: a feasibility study

PLOS ONE

Dear Dr. Lai,

Thank you for submitting your manuscript to PLOS ONE. After careful consideration, we feel that it has merit but does not fully meet PLOS ONE’s publication criteria as it currently stands. Therefore, we invite you to submit a revised version of the manuscript that addresses the points raised during the review process.

Please go through the reviewer comments and revise the manuscript indicating how each point raised by the reviewers has been addressed.

We would appreciate receiving your revised manuscript by Apr 25 2020 11:59PM. To enhance the reproducibility of your results, we recommend that if applicable you deposit your laboratory protocols in protocols.io, where a protocol can be assigned its own identifier (DOI) such that it can be cited independently in the future. For instructions see: http://journals.plos.org/plosone/s/submission-guidelines#loc-laboratory-protocols

We look forward to receiving your revised manuscript.

Kind regards,

Iddya Karunasagar

Academic Editor

PLOS ONE

Journal Requirements:

Additional Editor Comments (if provided):

The reviewers have raised major concerns in the manuscript which needs to be addressed through a major revision addressing all concerns point by point.

Reviewers' comments:

Reviewer's Responses to Questions

**Comments to the Author**

1. Is the manuscript technically sound, and do the data support the conclusions?

Reviewer #1: Partly

2. Has the statistical analysis been performed appropriately and rigorously? 

Reviewer #1: No

3. Have the authors made all data underlying the findings in their manuscript fully available?

Reviewer #1: Yes

4. Is the manuscript presented in an intelligible fashion and written in standard English?

Reviewer #1: Yes

5. Review Comments to the Author

Reviewer #1: Weil et al., provide a manuscript detailing the feasibility of conducting a delayed intervention trial with the objective of measuring the patterns of fecal AMR genes in chicken farmers before and after poultry introduction to a “farm” or household. To my knowledge, the application of a delayed intervention RCT design for a longitudinal study of antimicrobial resistance maker gene prevalence in humans and livestock is novel. This feasibility study identifies many factors that should be considered in design and implementation of such a study. Many of the critical factors identified are omitted in this study, and some additional factors in data analysis are not mentioned. To my knowledge, using a Jaccard dissimilarity index is a novel approach to estimating shared antimicrobial resistance markers in populations or groups.

Major concerns:

1. The objective or hypothesis of the study is unclear. Please revise the sentence at the end of the introduction section to provide a more comprehensive description of study goals or objective(s) or any specific hypothesis.

2. The unbalanced design by not sampling the chickens at the time of introduction in both the intervention and control group, and not sampling humans at the same time as chickens is a major limitation and should motivate a more conservative interpretation of the results. While you recognize this limitation in the discussion, in my opinion it seems biologically unjustified to test associations between the resistance genes found in the human groups and the chickens provided to the control group 6 months after the human samples were collected. As you note in the introduction temporal aspects and history of exposure are critical. Further, the study design seems to assume a single chicken sample, taken 6 months after the human one year samples, is a realistic representation of the chicken samples provided to the intervention group. Previous animal (and human) studies have shown resistance markers change over time and especially in relation to antibiotic exposure (e.g. tetracycline exposure in the brooders) and life cycle or management changes.

3. In my opinion, there are too many places in the manuscript where you imply possible chicken to human transmission, including in the approach to analysis, the presentation of results, the interpretation/discussion, and the conclusions. Is this an objective of the study? (e.g. lines 48, 172, 254) Please refine the description of the study objectives and in doing so justify the approach. Specifically comment on why you did not collect base-line and one year samples from the poultry prior to and after introduction to the households in both groups, and why you did not collect 18 month samples in humans. Please refine the description in the methods describing the duration of contact between humans and poultry at the 18 month time point prior to chicken sample collection.

4. An important aspect of the feasibility analysis would be an analysis of the variability in gene marker occurrence in the control and intervention groups over time and between subjects within groups. Please explore this variability using the data available and describe how such variability would influence future study design (e.g. power calculations to explore effect size estimates). While you have commented on the small sample size limitation in the discussion, you did not specifically comment on the infrequent repeated measures and evaluating variability in outcomes over time and power needed to discriminate between noise (random variation) and true trends over time. In my opinion, 2 time points are insufficient to make inferences regarding trends over time. Longer term surveys of antimicrobial resistance trends demonstrate sample frequency influences interpretation.

5. Because it appears you did not account for within subject repeated measures, nor did you appear to account for clustering (the hierarchical nature of subjects and chickens within households and within villages), in the discussion, please comment on: 1) longitudinal studies that include repeated measures within subjects should include assessments of correlation within subjects and models assessing associations should account for repeated measures and autocorrelation, 2) the potential effect of clustering within households and villages (e.g. humans in the same villages (or connected households within villages) may have greater resistance gene similarity compared to humans from other villages, independent of the chicken exposure treatment). For example, inclusion criteria for this study was the household must have at least 1 child under the age of 5. What if the household also had school-aged children who attend school? This is just one obvious example (of many) that creates a possible contact network between households (not to mention within village shared water sources, which you mentioned, or shared latrines or shared foods or shared food production environments).

6. Building on the previous comment, how do you control for the possibility of community acquired resistance genes in this study? Please comment on the limitations in this study design? Does this preliminary work suggest it would be possible to replicate the study across multiple villages? A challenge with longitudinal infectious disease studies is when intervention and control groups co-mingle and you cannot control for the possible contact and transmission between the intervention and control groups. This issue should be addressed in additional analyses of the data and in the discussion; for example the human baseline resistance gene profiles in the control group (or possibly a pooled baseline of the time zero control and intervention groups) could be used as the centroid to test the similarity between intervention or control subjects at one year or chickens at 18 months compared to controls (or pooled data) at baseline. I would recommend conducting a similarity analysis to first demonstrate that the baselines in the 2 human groups are not different (not sure how one might do tis in the absence of replication), before pooling the baseline data for subsequent analysis. Given that 10 of the 32 markers were found in only chickens and 13 of the 32 markers were found in both chickens and humans, and the majority of the markers 22 were found in humans, I suspect there may be greater similarity in markers found in the one year samples compare to base-line humans than in markers found in chickens compared to base-line humans. And there may be an issue with the uneven frequency of sampling (i.e. lower overall sample numbers in the chickens results in a under-estimate of the true frequency of resistance markers in this population.)

7. Methods – please add more details on the methods of recruitment of the women in this study as a subset of the original RCT. Was this a convenience sample subset of the original eligible subjects? Please add details as to whether stool sample collection was included in the parent RTC plan, or if the sample collection and outcome measures in this feasibility study were a post-hoc additional to the original study.

8. Please add some discussion (at least) and possibly some analysis of the Jaccard dissimilarity index approach to estimating shared antimicrobial resistance markers in populations or groups. Are there other studies that have applied this approach? Are there any studies that have compared using this summary index in a regression model with a more traditional approach using a binary outcome such as presence or absence of a gene marker or class of genes with corrections for multiple comparisons?

9. Are chickens ever free-ranging during the course of the study? How long was it until slaughter for the chickens provided to the intervention group? Were slaughtered chickens sold in the village? How frequently were intervention household chickens purchased by (or shared with) control households? My experience with poultry introduction programs in rural communities which provided coup construction in addition to animals, is that the poultry are often raised extensively, (i.e. allowed to forage outside the coup areas) for significant periods of time during the day or their lifetime, and may be returned to owner’s coups at night. Chickens will then forage in cattle (and other livestock) dung and the broad environment (including human waste and especially any human food waste such as vegetable scraps which may have fecal/manure contaminants and be a source of resistance genes). Thus the chicken diet and potential exposure is not well controlled in many rural systems, and chickens may simply be a sentinel for common environmental exposure risk. Were these variables considered and controlled for in this study? If appropriate this issues should be addressed in the discussion section.

10. In my opinion, the authors over emphasize speculative transmission (based on similarity) of resistance markers between humans and chickens, when alternative transmission dynamics, independent of poultry contact, such as human to human or human to environment to human, can explain the observed data and are not explored in the analysis – if the intervention subjects and the control subjects are considered to belong to the same population (i.e. from the same village) then at base-line almost all of the observed genes present in the human population are also found in either the intervention or control group at 12 months. From table 2 and table S1, it appears 4 (12%) of 32 (did I count correctly?) total genes were found in either the intervention or control groups at 1 year [aaC2, AAC(6)-Ib-cr, CFE-1 and LAT] but not at baseline, and none of these markers were found in chickens. It appears that the majority of genes found in chickens at 18 months were pre-existing in humans at baseline (i.e. before introduction of chickens for this trial), except for 10 genes (31% of all genes), [vanB, vanC, msrA, erm A, ermC, SHV(238S240E), SHV(238S240K), OXA-10 group, and OXA-58 group, QnrB-8 group], which were only found in chickens. There are many examples where a marker was found in one or both groups at one year but was not observed in chickens [AAC(6)-Ib-cr, aacC2, CTX-M-9 Group, SHV(156D), ACC-3, ACT 5/7 group, CFE-1, LAT, QnrB-1 group ] including at least 4 elements found in the control group prior to direct chicken contact. Further there are many examples were a resistance marker existed in baseline samples in one group and emerged in the other group at 12 months [CTX-M-1 Group existed in intervention at base then found in control at 1 year; SHV, SHV(238G240E), ACT 5/7 group, QnrB-5 group, QnrS in controls at base and found in intervention at 1 year]. Overall these findings suggest to me there is more similarity between human samples in the 2 groups at baseline and one year, than there is between human and poultry samples. While you discussed this concepts, with some specific examples in the discussion section. These associations should be explored more thoroughly.

Specific comments:

Line 34 – eliminate “growth promoters” – all antimicrobial use in livestock production likely contributes to emergence and selection of AMR genes in livestock and possibly subsequently to humans either through direct exposure to animals and their housing environments or through indirect exposure through the food chain (i.e. contaminated food products harvested from animals).

Line 34 and line 77 – delete “major” – this is subjective; I think many believe, and some research data would suggest, the major contributor to acquisition of AMR genes in humans is antimicrobial use in humans. See for example the systematic review and meta-analysis by Tang et al. (Lancet Planet Health 2017;

http://dx.doi.org/10.1016/S2542-5196(17)30141-9).

Line 37 – delete “from”

Line 47 – revise order, this is confusing as it does not represent what I interpret is the actual study design and sampling protocol… it might read “After one year of exposure to chickens, seven new AMR genes were detected in the intervention group at one year, while six new AMR genes were detected in the control group at one year and 6 months prior to when this group received chickens.”

Line 49 – revise “Women who had direct contact with the chickens sampled in the study had greater similarities in AMR resistance gene patterns to chickens than those who did not have direct contact with chickens sampled.” This sentence is misleading. In your model using the similarity outcome, at base-line (i.e. before any contact) and at one year (after contact), women with direct contact with chickens during the year (i.e. the intervention group) actually had lower similarity to chicken samples (taken at 18 months) than the control group (who had no direct household chicken contact at either baseline or one year).

Line 54 – revise - there were not chicken samples at both baseline and follow up time-points. The chicken samples are temporally distant from the human samples and it is not clear the length of contact time between humans and chickens before the chicken samples were taken at 18 months .

Line 68 – “The problem is accentuated…” The information in this sentence seems logical, but without data, is speculative – please revise this sentence. Please clarify, is this a hypothesis statement for future studies or can you provide a reference supporting this statement with empirical data?

Line 93 – please end the introduction with a sentence describing the actual objective(s) (or hypothesis?) of the study – was the objective simply to determine the feasibility of conducting a longitudinal study of AMR gene patterns in chicken farmers and their poultry in a delayed intervention RCT? Or was the objective to test a specific hypothesis? From the current sentence, it seems the objective was to determine patterns of AMR genes in stool of chicken farmers before and after poultry introduction, but the methods appear to go beyond this descriptive objective to test associations between resistant elements found in poultry and their caregivers. Yet, as discussed under general concerns, the study design for this test of association is incomplete and the statistical approach ignores the hierarchical nature (clustering) and repeated measures (correlation) in the sample populations.

Line 107 - What was the duration of the brooding period, and more specifically what was the duration of feeding tetracycline-containing dietary supplements? Were these supplements provided in water or in solid feed? What was the target dose of tetracycline as fed per kilogram animal unit?

Line 110 – how are the survey data being used in this study? In particular the records of antimicrobial use for the women study participants and for treatment of all livestock on the farms. Was this a new survey developed only for this feasibility study, or were data extracted from a larger survey developed for the original larger RTC. Please provide a copy of the survey tool in the supplemental data, or provide a link to a permanent archive for the survey tool.

Line 113 – modify figure S1 to indicate the actual number of participants sampled in the 2 arms of the this feasibility study

Line 115 – so at 18 months, when the control group receives their first batch of brooding chicks (which were receiving medicated feed or feed supplements at the time of sampling?) you take chicken fecal samples. This seems to be an apples and oranges comparison group, because you have longitudinal samples from the humans and a single point in time sample from the chicks during a time when the chicks appear to be receiving tetracycline. Also the chicks are introduced to the control households and chicken samples are taken 6 months after the last human samples with no description of the duration of direct contact between chickens and humans in the control group, prior to sampling the chickens. How long were the chicks exposed to the humans and the farm environment before they were sampled? What relevance does the antibiotic resistance gene screening in the poultry at 18 months have toward the study objective? This might be the place in methods section to provide more detail that informs these concerns.

Line 154 – what is meant by environmental contaminants? These samples appear to be controlling for contaminants introduced within the laboratory setting? There would be other more appropriate controls for contaminants introduced during sampling in the farm or household environments (i.e. sham samples collected on farm). Perhaps revise the wording to be more specific.

Line 173 – revise sentence, there are no “baseline” chicken samples

Line 254 – again “both at baseline and follow-up” is modifying chicken samples, when there was only one time point for chicken samples

Line 291 – add something to the effect of, “and people who did not care for chickens during the 12 months of human sampling (control group) harbored many of the same genes at one year.”

Line 294 – revise – this wording is confusing, the humans who had direct contact with chickens at one year in the study when samples were taken was the intervention group, correct? So the control group had no direct contact with chickens at one year, yet their samples were more similar to chicken samples collected at 18 months. Was this expected? What are possible explanations for this finding?

Line 295 – delete or revise “AMR gene patterns in both groups became more similar to AMR…” the use of the word “became” suggests a temporal association that could be perceived to imply chicken gene patterns are the driver of human gene patterns. I suggest, this sentence could also be written as chicken gene patterns at 18 months were more similar to human patterns at one year than to human patterns at base-line. Either delete or revise after analyzing the similarity of one year in both human groups to human base-line. Highlighting this comparison demonstrates the limitation of not having longitudinal chicken samples.

Line 303 – revise, change to “to humans or animals” – there are examples where humans serve as a reservoir of resistant organisms that appear to be transmitted to animals, further antibiotic resistance is also detrimental to animal health

Line 308 – delete or revise – “Additionally, we observed a high rate of AMR genes overall in humans and in chickens in rural Uganda” what is the comparison group – please provide a reference - higher than what other group? Urban humans and chickens in Uganda? Rural humans and chickens in other countries?

Line 366 – please add another limitation; the number of replicates over time (in both humans and chickens) are limited and do not represent a diversity of timing relative to chicken production cycle and antibiotic use in that cycle.

Line 386 – “significant changes” – how do you define “significant” here? This conclusion can be kept if the recommended analysis comparing 1 year human to base-line human demonstrates a significant change in prevalence (frequency) – otherwise change wording to “we identified differences in the 2 time point analyzed” Of course you also identified many (more?) similarities between the 2 time points. And also there were some subjects where markers were found at baseline and not at one year. Highlight the challenges of modeling the dynamics at the level of subjects within treatment group.

Figure 2 – legend – source chicken crtl should be 18 months

In the reference section, organism genus species names should be written in italics

6. PLOS authors have the option to publish the peer review history of their article (what does this mean?). If published, this will include your full peer review and any attached files.

Reviewer #1: No

---

## [Author Response · Author response to Decision Letter 0]

23 Apr 2020

Please see uploaded word document with overall changes in the revised manuscript and point by point response.

---

## [Decision Letter · Decision Letter 1]

20 May 2020

Gut carriage of antimicrobial resistance genes in women exposed to small-scale poultry farms in rural Uganda: a feasibility study

PONE-D-20-03262R1

Dear Dr. Lai,

We are pleased to inform you that your manuscript has been judged scientifically suitable for publication and will be formally accepted for publication once it complies with all outstanding technical requirements.

With kind regards,

Iddya Karunasagar

Academic Editor

PLOS ONE

Additional Editor Comments (optional):

All comments addressed satisfactorily

Reviewers' comments:

Reviewer's Responses to Questions

**Comments to the Author**

1. If the authors have adequately addressed your comments raised in a previous round of review and you feel that this manuscript is now acceptable for publication, you may indicate that here to bypass the “Comments to the Author” section, enter your conflict of interest statement in the “Confidential to Editor” section, and submit your "Accept" recommendation.

Reviewer #1: (No Response)

2. Is the manuscript technically sound, and do the data support the conclusions?

Reviewer #1: Yes

3. Has the statistical analysis been performed appropriately and rigorously? 

Reviewer #1: Yes

4. Have the authors made all data underlying the findings in their manuscript fully available?

Reviewer #1: Yes

5. Is the manuscript presented in an intelligible fashion and written in standard English?

Reviewer #1: Yes

6. Review Comments to the Author

Reviewer #1: Drs. Weil, Lai and other study authors,

Thank you for the attention to detail in addressing my suggestions and concerns in the revised manuscript. I have no additional requests for revisions, only minor comments for your consideration. I truly enjoyed reviewing your manuscript and I am glad you believe my comments were constructive. These additional four comments reference the line numbers in the revised document without the tracked changes.

Line 67 – consider adding an additional factor, "insufficient sanitation and hygiene infrastructure for managing human and animal waste."

Line 87 - either here, or later in the discussion, you might consider recognizing one of the classic (seminal?) studies conducted by Stewart Levy

Changes in intestinal flora of farm personnel after introduction of a tetracycline-supplemented feed on a farm. Levy SB FitzGerald GB Macone AB. N Engl J Med 1976 09 Sep;295(11):583-8 PMID:950974

Emergence of antibiotic-resistant bacteria in the intestinal flora of farm inhabitants. Levy SB. J Infect Dis 1978 May;137(5):689-90 PMID:351088

Line 278 – I have always been taught to avoid the concept of a “trend toward a difference” based on a marginal p value, but I suppose this is personal preference and I am not going to be pedantic and push for a change in the language, as I believe it is reasonable in a pilot or feasibility study an association at P<0.1 might be worth recognizing as of interest for future study

Line 298 – typo “uwed”

7. PLOS authors have the option to publish the peer review history of their article (what does this mean?). If published, this will include your full peer review and any attached files.

Reviewer #1: No

---

## [Editor Report · Acceptance letter]

1 Jun 2020

PONE-D-20-03262R1 

Gut carriage of antimicrobial resistance genes in women exposed to small-scale poultry farms in rural Uganda: a feasibility study 

Dear Dr. Lai:

I am pleased to inform you that your manuscript has been deemed suitable for publication in PLOS ONE. Congratulations! Your manuscript is now with our production department. 

With kind regards,

on behalf of

Dr. Iddya Karunasagar 

Academic Editor

PLOS ONE